# DRACO: A Denoising-Reconstruction Autoencoder for Cryo-EM

**Yingjun Shen**[1,2][*]  **Haizhao Dai**[1,2][*]

**Qihe Chen**[1,2]  **Yan Zeng**[1,2]  **Jiakai Zhang**[1,2]  **Yuan Pei**[1,3]  **Jingyi Yu**[1]

[1]School of Information Science and Technology, ShanghaiTech University.
[2]Cellverse Co, Ltd.
[3]iHuman Institute, ShanghaiTech University.
{shenyj2022,daihzh2023}@shanghaitech.edu.cn
{chenqh2024,zengyan2024,zhangjk,peiyuan,yujingyi}@shanghaitech.edu.cn

## Abstract

Foundation models in computer vision have demonstrated exceptional performance in zero-shot and few-shot tasks by extracting multi-purpose features from large-scale datasets through self-supervised pre-training methods. However, these models often overlook the severe corruption in cryogenic electron microscopy (cryo-EM) images by high-level noises. We introduce DRACO, a Denoising-Reconstruction Autoencoder for CryO-EM, inspired by the Noise2Noise (N2N) approach. By processing cryo-EM movies into odd and even images and treating them as independent noisy observations, we apply a denoising-reconstruction hybrid training scheme. We mask both images to create denoising and reconstruction tasks. For DRACO's pre-training, the quality of the dataset is essential, we hence build a high-quality, diverse dataset from an uncurated public database, including over 270,000 movies or micrographs. After pre-training, DRACO naturally serves as a generalizable cryo-EM image denoiser and a foundation model for various cryo-EM downstream tasks. DRACO demonstrates the best performance in denoising, micrograph curation, and particle picking tasks compared to state-of-the-art baselines.

## 1 Introduction

Foundation models in computer vision have demonstrated remarkable capabilities in zero-shot and few-shot tasks. These models learn to extract multi-purpose visual features from large-scale, diverse datasets through text-guided [1, 2, 3] or self-supervised [4, 5] pre-training methods such as masked image modeling (MIM) [6]. The features can then be applied to various downstream tasks. For instance, DINOv2 [5] is trained on a large-scale curated dataset and shows significant performance improvements in classification, retrieval, segmentation, etc. The success of vision foundation models has stimulated advances across various scientific disciplines. Due to the diverse modalities of scientific imaging, training domain-specific foundation models [7, 8, 9, 10] is essential to meet specific demands. For example, the UNI [7] foundation model for tissue imaging is pre-trained on 100 million images for 34 representative clinical downstream tasks.

In structural biology, cryogenic electron microscopy (cryo-EM) stands as a pivotal bio-imaging technique [11]. Unlike optical imaging methods, cryo-EM possesses several distinctive characteristics: first, cryo-EM utilizes high-energy electron beams as its illumination source [12] and direct detector

---

[*]The authors contributed equally to this work.

38th Conference on Neural Information Processing Systems (NeurIPS 2024).

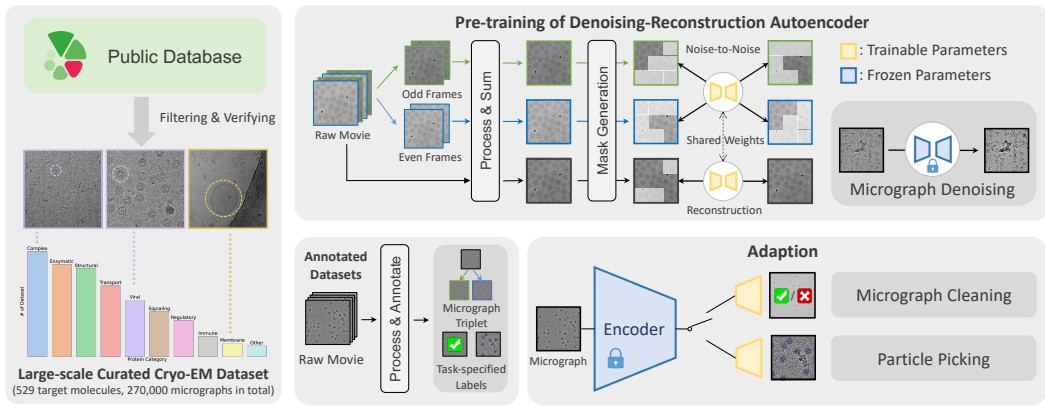

Figure 1: **Overview of DRACO.** For pre-training, we construct a large-scale curated dataset containing 529 types of protein data with over 270,000 cryo-EM movies or micrographs. Based on this, we present DRACO, a denoising-reconstruction autoencoder for cryo-EM. A pre-trained DRACO naturally serves as a generalizable cryo-EM image denoiser and a foundation for various downstream model adaptions such as micrograph curation and particle picking.

device (DDD) captures a continuous multi-frame sequence, often called a *movie* [13]. To mitigate specimen damage during exposure, the electron dose per frame is restricted, leading to *extremely low signal-to-noise ratios (SNR)* in the captured images. Second, cryo-EM employs motion correction to counteract blurring induced by specimen drift during exposure to ultimately obtain a sharper single micrograph [13]. Last, the acquired images comprise hundreds of thousands of target protein particles with diverse poses. To resolve 3D structures, researchers utilize a pipeline composed of multiple downstream tasks including micrograph curation, micrograph denoising, particle picking, pose estimation, and ultimately high-resolution reconstruction.

In line with denoising autoencoders [14], which consider "robustness to partial destruction of the input" a good representation criterion, existing self-supervised learning methods, such as masked autoencoders (MAE) [6], have been successful in learning expressive representations by reconstructing the missing patches from a partially masked image. However, in cryo-EM, these methods overlook the severe corruption caused by pixel-level random noise, leading to degraded performance. To be more robust to noises, DMAE [15] reconstructs the clean image from the masked one that is further corrupted by synthetic Gaussian noise. Nevertheless, cryo-EM clean reference images are impossible to obtain due to the fragile biological specimens, posing a significant challenge.

In this paper, we present DRACO, a Denoising-Reconstruction Autoencoder for CryO-EM, as shown in Figure 1. Inspired by Noise2Noise (N2N) [16], which learns to denoise images using only paired noisy images, we divide the original movie into two sub-movies based on odd and even frame numbers, processing them into odd and even images. We treat them as two independent, noisy observations of the underlying true signal, thus the idea of N2N can apply. During training, we partially mask both images, creating masked regions and unmasked regions corresponding to denoising and reconstruction tasks: in the unmasked region, the odd noisy patch learns to recover the even noisy patch, and vice versa. In the masked region, we introduce relatively low-noise images from the complete movie to supervise the reconstructed results. This denoising-reconstruction hybrid training scheme achieves the robust feature extraction of noisy cryo-EM images.

The quality of training samples is crucial for the general-purpose feature extraction of cryo-EM images. Direct access to the public database leads to varying data quality, inconsistent data formats, or missing annotations. Therefore, we construct a large-scale, high-quality, and diverse single-particle cryo-EM image dataset by curating and manually processing 529 sets of data from EMPIAR [17], obtaining over 270,000 cryo-EM movies or micrographs in total. After pre-training, DRACO naturally serves as a generalizable cryo-EM image denoiser and a foundation for various downstream model adaptions. We hence explore the performance of DRACO on three downstream tasks: micrograph curation, denoising, and particle picking. Extensive experiments show that DRACO outperforms the state-of-the-art baselines in all downstream tasks. We will release code, pre-trained/fine-tuned models, and the large-scale curated dataset.

## 2 Related Work

Our work aims to extend the vision foundation model to the field of cryo-EM. We therefore only discuss the most relevant works in respective fields.

**Vision foundation models in computer vision.** Vision foundation models are pre-trained on large-scale image datasets [18, 19] using self-supervised learning methods [20], aimed at extracting general visual signals rapidly adaptable to various downstream visual tasks [21, 22, 23, 24]. Techniques for pre-training vision foundation models, such as contrastive learning [25, 26, 1, 27] and self-distillation [28, 5], focus on aligning features across different models or modalities, while another method, masked image modeling [6, 29, 30, 4], reconstructs features from masked images to capture high-level visual semantics. However, these existing vision foundation models are not directly applicable to cryo-EM imaging. In particular, their application to cryo-EM imaging is limited by the high noise levels in micrographs, which degrade signal capture. Therefore, we propose a denoising-reconstruction pre-training framework that is robust to highly noisy cryo-EM micrographs, making it suitable for specific downstream tasks in cryo-EM.

**Vision foundation models in life science.** The remarkable success of foundation models has extended to various life science imaging domains, including applications in retinal [10], fluorescence microscopy [9], histopathology [7, 8], and radiology imaging [31]. These models have shown considerable effectiveness in tasks such as disease diagnosis, lesion detection, and image restoration within these fields. In contrast to these domains, which benefit from extensive and well-curated datasets supporting model training, the cryo-EM field lacks such resources. To fill this gap, we have developed a well-curated, large-scale dataset specifically designed to support the training of cryo-EM foundation models, ensuring that they can be effectively applied in this specialized field.

**Cryo-EM image denoising.** To tackle the issues of low SNR and complex noise patterns in cryo-EM images, traditional denoising techniques often employ noise models like the Poisson-Gaussian model [12, 32] and rely on filtering methods [33, 34, 35] to denoise. However, these methods oversimplify the noise patterns, which can lead to the loss of high-frequency signal details. Recently, NT2C [36] uses Generative Adversarial Network to learn the noise patterns for denoising, but it requires simulated datasets as the clean references. Another series of learning-based methods [37, 38] make full use of multi-frame data by generating odd and even images for denoising based on Noise2Noise (N2N) [16, 39] framework. These methods do not require clean images for denoising, but they still suffer from small-scale datasets and network architectures, which limit their generalizability. In this paper, we propose DRACO, pre-trained on a large-scale curated dataset, that can naturally serve as a generalizable denoiser for cryo-EM micrographs.

**Downstream tasks in single particle analysis.** An effective foundational model can benefit downstream tasks in cryo-EM, including micrograph curation and particle picking. Micrograph curation aims to ensure that only high-quality images are selected for further analysis, yet current methods rely heavily on manual inspection [40, 41]. Particle picking involves identifying and extracting representative particles from micrographs, which is a critical task in the cryo-EM single particle analysis (SPA) reconstruction pipeline. Traditional methods [42, 43, 44], such as template matching [45, 46] and difference of Gaussians (DoG) method [47], heavily rely on prior information and require substantial *ad hoc* post-processing. Learning-based models [48, 49, 50], such as Topaz-Picking [51], crYOLO [52], and CryoTransformer [53], offer more streamlined processes but still face challenges in generalizability due to the limited data scale. Our DRACO, pre-trained on large-scale cryo-EM image datasets, can effectively adapt to these tasks and demonstrate strong generalization capabilities.

## 3 Preliminary: Imaging Formation Model

Cryo-EM uses a Direct Detector Device (DDD) camera [32] for their notably higher detective quantum efficiency (DQE) compared to traditional cameras. This allows recording the micrograph as a multi-frame movie rather than a single integrated exposure. In this setup, a movie is a series of continuous multi-frame images, denoted as $\mathcal{M} = \{\hat{I}_i\}_{i=1}^{M}$, where each frame $\hat{I}_i$ is an independent observation of the true signal $I$.

Ideally, the imaging process in cryo-EM involves two main steps: 1) projecting the 3D density volume of the region of interest $V(x, y, z) : \mathbb{R}^3 \to \mathbb{R}$ along the $z$-axis via weak-phase object approximation [54], and 2) modulating the projection image with the Point Spread Function (PSF) $g$ of the cryo-EM

optical lens, expressed as:

$$I = g * \int V(x, y, z) \, \mathrm{d}z. \tag{1}$$

where $I$ is considered as the true signal.

However, in practice, the captured frames suffer from extremely low SNR due to the limited electron dosage and the high sensitivity of DDD. The main noise source is Poisson (shot) noise from the detector, denoted as Poisson($I$), arising from the inherent uncertainty of the electron measurement [12]. We assume that the additional noise types like heat, readout, and dark current noise are collectively modeled as a zero-mean Gaussian noise $\mathcal{G}$ with an unknown variance $\sigma^2$ [42]:

$$\hat{I} = \mathrm{Poisson}(I) + \mathcal{G}. \tag{2}$$

As the number of observations increases, the average of these movie frames converges to the true signal:

$$\mathbb{E}[\hat{I}] = I \approx \frac{1}{M} \sum_{i=1}^{M} \hat{I}_i. \tag{3}$$

We define the image noise $\epsilon$ as the difference between the captured and clean images for each frame. Thus, the expectation (mean value) of noise distribution is:

$$\mathbb{E}[\epsilon] = \mathbb{E}[\hat{I}_i - I] = 0. \tag{4}$$

Thus, we derive that the cryo-EM image noise is zero-mean. This conclusion is also aligned with existing cryo-EM reconstruction methods [42, 41], which directly assume that the noise distribution is an additive zero-mean Gaussian yet can still achieve high-resolution reconstruction. We are well aware that this derivation is relatively trivial compared to an actual theoretical analysis [12], but this gives us intuitive guidance to integrate the N2N [16] idea: learning to denoise images from solely noisy image pairs, into our pre-training framework.

**Movie to micrograph triplets.** Given an $M$-frame movie $\mathcal{M}$, we divide it into odd frames $\mathcal{M}^{\mathrm{o}} = \{I_{2i-1}\}_{i=1}^{\lceil M/2 \rceil}$ and even frames $\mathcal{M}^{\mathrm{e}} = \{I_{2i}\}_{i=1}^{\lfloor M/2 \rfloor}$. An off-the-shelf motion correction method [13] is then applied to correct cross-frame drifts in $\mathcal{M}$, $\mathcal{M}^{\mathrm{o}}$, and $\mathcal{M}^{\mathrm{e}}$. By summing up the frames within each subset, we generate three micrographs with the same shape: the original micrograph $\hat{\mathbf{I}}$, odd micrograph $\hat{\mathbf{I}}^{\mathrm{o}}$, and even micrograph $\hat{\mathbf{I}}^{\mathrm{e}}$, As aforementioned, all these micrographs are expected to reflect the true signal $I$ but corrupted by noises.

## 4 Denoising-reconstruction Autoencoder

We introduce DRACO, a denoising-reconstruction autoencoder for cryo-EM, as illustrated in Figure 2. Different from existing masked imaging modeling methods, our model uses paired odd-even micrographs as inputs for the denoising target on visible patches. Further, we utilize the original micrographs as the additional supervision signal for reconstruction on masked patches.

**Masking.** Following the standard scheme in Vision Transformer (ViT) [55], each micrograph in a triplet, consisting of one original, one odd, and one even micrograph from the same movie, is divided into regular non-overlapping patches. We create patch sets $\{\mathbf{x}_i^{\mathrm{o}}\}_{i=1}^{N}$ for the odd, $\{\mathbf{x}_i^{\mathrm{e}}\}_{i=1}^{N}$ for the even, and $\{\mathbf{x}_i\}_{i=1}^{N}$ for the original micrograph, where $N$ represents the number of patches. For the odd and the even micrographs used as inputs in our model, we generate two sets of binary masks, $\{\mathbf{m}_i^{\mathrm{o}}\}_{i=1}^{N}$ and $\{\mathbf{m}_i^{\mathrm{e}}\}_{i=1}^{N}$, with a mask ratio $\gamma$. Here, $\mathbf{m}_i = 1$ means the $i$-th patch is masked, and $0$ means unmasked. Additionally, we ensure that a patch can be 1) only visible in one of them, or 2) masked in both. This strategy ensures that each visible patch has no information sharing of its corresponding patch on the other micrograph. Notably, this requires $\gamma \geq 0.5$ for each input micrograph.

**Network architecture.** For DRACO's pre-training, we employ a ViT-based encoder-decoder architecture following the MAE framework [6]. Positional embeddings are first added to the input patches, which are then masked to select only visible patches, denoted as $\{\mathbf{v}_i\}_{i=1}^{\gamma N}$. The encoder, $G_{\mathrm{enc}}$, is a ViT that transforms these visible patches from either odd or even micrographs into latent features. To

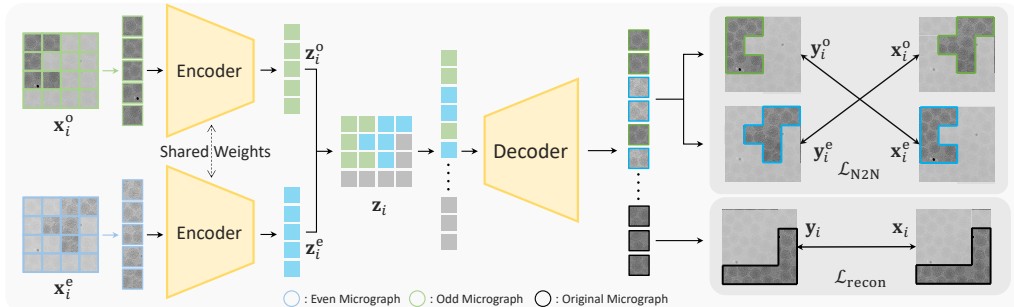

Figure 2: **The pipeline of DRACO.** Given a pair of partially masked odd and even micrographs, the encoder takes odd-visible patches and even-visible patches as inputs. The unmasked latent patches are combined with masked latent patches together to generate the latent representation $\mathbf{z}_i$. Then the latent representation passes through the decoder to generate predicted patches. The N2N loss is applied to odd-visible predicted patches with corresponding even input patches, and vice versa. The reconstruction loss is applied to both invisible predicted patches with higher SNR input patches.

align with the original unmasked size, zeros are padded to the latent features based on the positions indicated by the corresponding masks before the encoder outputs them:

$$\{\mathbf{z}_i^{\mathrm{o}}\}_{i=1}^N = G_{\mathrm{enc}}\left(\{\mathbf{v}_i^{\mathrm{o}}\}_{i=1}^{\gamma N}, \{\mathbf{m}_i^{\mathrm{o}}\}_{i=1}^N; \theta_{\mathrm{enc}}\right), \quad \{\mathbf{z}_i^{\mathrm{e}}\}_{i=1}^N = G_{\mathrm{enc}}\left(\{\mathbf{v}_i^{\mathrm{e}}\}_{i=1}^{\gamma N}, \{\mathbf{m}_i^{\mathrm{e}}\}_{i=1}^N; \theta_{\mathrm{enc}}\right). \quad (5)$$

Each of the latent features from odd and even micrographs retain part of the original micrograph's information. To reconstruct the micrograph, we generate the latent representation $\{\mathbf{z}_i\}_{i=1}^N$ for further processing by the decoder:

$$\mathbf{z}_i = (1 - \mathbf{m}_i^{\mathrm{o}}) \cdot \mathbf{z}_i^{\mathrm{o}} + (1 - \mathbf{m}_i^{\mathrm{e}}) \cdot \mathbf{z}_i^{\mathrm{e}} + \mathbf{m}_i^{\mathrm{o}} \cdot \mathbf{m}_i^{\mathrm{e}} \cdot [\mathtt{MASK}], \quad (6)$$

where the $[\mathtt{MASK}]$ token is a shared learnable embedding representing the masked patch to be predicted. Finally, the decoder $G_{\mathrm{dec}}$ takes the latent representation $\{\mathbf{z}_i\}_{i=1}^N$ with another positional embedding added as input and predicts all masked and visible patches to reconstruct the complete micrograph:

$$\{\mathbf{y}_i\}_{i=1}^N = G_{\mathrm{dec}}(\{\mathbf{z}_i\}_{i=1}^N; \theta_{\mathrm{dec}}). \quad (7)$$

**Denoising target.** Inspired by N2N and Topaz-Denoise [37], which predict denoised images only from paired noisy images, we introduce an image-denoising target on visible patches. For any patch $\mathbf{x}_i$ visible in only the odd micrographs, the model predicts its counterpart in the even micrographs, and vice versa, as illustrated in Figure 2. As mentioned earlier, the expectation (mean) values of the odd and even micrographs are the true signal. Therefore, following Topaz-Denoise, we employ a patch-wise L2 loss function for each visible patch, aiming for regressing the true signal:

$$\mathcal{L}_{\mathrm{N2N}}(\mathbf{x}_i^{\mathrm{o}}, \mathbf{x}_i^{\mathrm{e}}, \mathbf{y}_i^{\mathrm{o}}, \mathbf{y}_i^{\mathrm{e}}) = \begin{cases} \|\mathbf{x}_i^{\mathrm{e}} - \mathbf{y}_i^{\mathrm{o}}\|_2^2, & \text{if } \mathbf{m}_i^{\mathrm{o}} = 0, \\ \|\mathbf{x}_i^{\mathrm{o}} - \mathbf{y}_i^{\mathrm{e}}\|_2^2, & \text{if } \mathbf{m}_i^{\mathrm{e}} = 0. \end{cases} \quad (8)$$

**Reconstruction target.** For masked patches, we let the decoder predict the pixel value of patches from the original micrograph with higher SNR compared to odd and even micrographs for better reconstruction quality. Thus, the reconstruction loss is:

$$\mathcal{L}_{\mathrm{recon}}(\mathbf{x}_i, \mathbf{y}_i) = \|\mathbf{x}_i - \mathbf{y}_i\|_2^2, \quad \text{if } \mathbf{m}_i^o \cdot \mathbf{m}_i^e = 1. \quad (9)$$

**Training objective.** During training, we combine the N2N loss with the reconstruction loss:

$$\mathcal{L} = \mathcal{L}_{\mathrm{N2N}} + \lambda \mathcal{L}_{\mathrm{recon}}, \quad (10)$$

where $\lambda$ is a hyper-parameter set to $1.0$ in all our experiments.

# 5 Experiments

**Large-scale curated dataset for pre-training.** The effectiveness and robustness of DRACO depend heavily on the quality and scale of the cryo-EM pre-training dataset. However, direct access to public databases like the Electron Microscopy Public Image Archive (EMPIAR) [17] results in variations in data quality, inconsistent data formats, and inaccurate or even missing annotations. To overcome these challenges, we have developed a data generation workflow. First, we select datasets with reported resolutions better than 10 Å, ensuring high-quality data acquisition. Next, we collect the raw data, including metadata, movies, and micrographs, from pre-defined high-quality datasets available on EMPIAR. Finally, we re-process the raw data using cryoSPARC [41] through a custom processing pipeline designed to exclude low-quality micrographs and movies, generate annotations for downstream tasks, and verify the resolutions of the reconstructed results. This workflow has allowed us to compile a large-scale, curated cryo-EM dataset containing over 270,000 raw micrographs and more than 50,000 raw movies from 529 verified single-particle cryo-EM datasets, occupying approximately 25 TB of disk storage in total. Details of the cryoSPARC processing pipeline are provided in Appendix C.1.

**Data augmentation.** Each micrograph in a triplet goes through the same data augmentation process: randomly cropping to between 1/16 and 1/4 of the original size (typically $4096 \times 4096$), resizing to $256 \times 256$, applying random horizontal and vertical flips, and finally normalizing based on the mean and standard deviation computed from the original micrograph. We randomly crop each micrograph 16 times within a single epoch for fully utilization.

**Pre-training details.** We explore the performance of DRACO using two ViT architectures for the encoder, ViT-B and ViT-L, denoted as **DRACO-B** and **DRACO-L**, respectively. The decoder of DRACO uses 8 Transformer blocks with embedding dimension 512, followed by a three-layer convolution neck and a linear projection layer with an output dimension $16 \times 16$, which is also the patch size of the input. The mask ratio for the one input micrograph is 0.75 by default. To fully utilize our large-scale curated dataset, we warm up DRACO on the original 270,000 micrographs based on the MAE training scheme for 200 epochs. Then we adopt our novel denoising-reconstruction pre-training for 400 epochs. The warm-up stage takes 6 hours, and the pre-training stage takes 16 hours on a GPU cluster with 64 NVIDIA A800 GPUs, requiring approximately 80 GB of memory for a batch size of 4096.

## 5.1 Particle Picking

Particle picking aims to accurately locate particles in highly noisy micrographs, which is directly related to the resolution of the final reconstructed result. For adaption, we conduct supervised fine-tuning based on the ViT-based object detection framework Detectron2 [21] on our curated dataset. The results show that DRACO is capable of accurately detecting particles of various shapes and sizes across three challenging datasets.

**Dataset.** To create the annotated dataset for adaptation, we employ the standard workflow of cryoSPARC to generate a high-quality annotated dataset containing over 80,000 full micrographs and approximately 8 million particles across 46 types of protein. Detailed descriptions of the annotation workflow can be found in Appendix C.2. The test dataset includes three full micrograph sets with EMPIAR ID 1) 10081: Human HCN1 channel protein [56], the protein structure has a similar shape to ice, which could lead to false positive picking, 2) 10350: LetB transport protein [57], this kind of protein tends to aggregate together, posing challenges in accurate picking in crowded area, and 3) 10407: 70S ribosome [58], the micrographs are in the extremely low SNR.

**Baseline and metrics.** We compare DRACO with existing state-of-the-art learning-based methods for generalized particle picking, including **Topaz** [51], **crYOLO** [52], and **CryoTransformer** [53]. For ViT-based baselines, including **MAE**, **DRACO-B** and **DRACO-L** as aforementioned. We integrate them into Detectron2 framework by loading pre-trained weights of DRACO's encoder into Detectron2's encoder. Additionally, to show the effectiveness of pre-trained encoder, we compare with **Detectron2** trained from scratch. More configuration details can be found in Appendix D.1. We evaluate baselines in terms of conventional metrics including precision, recall, and F1 score, and the resolution of the resolved 3D structure from the picked particles, which is also a crucial metric. We process the particles selected by each method with the standard cryoSPARC workflow and finally

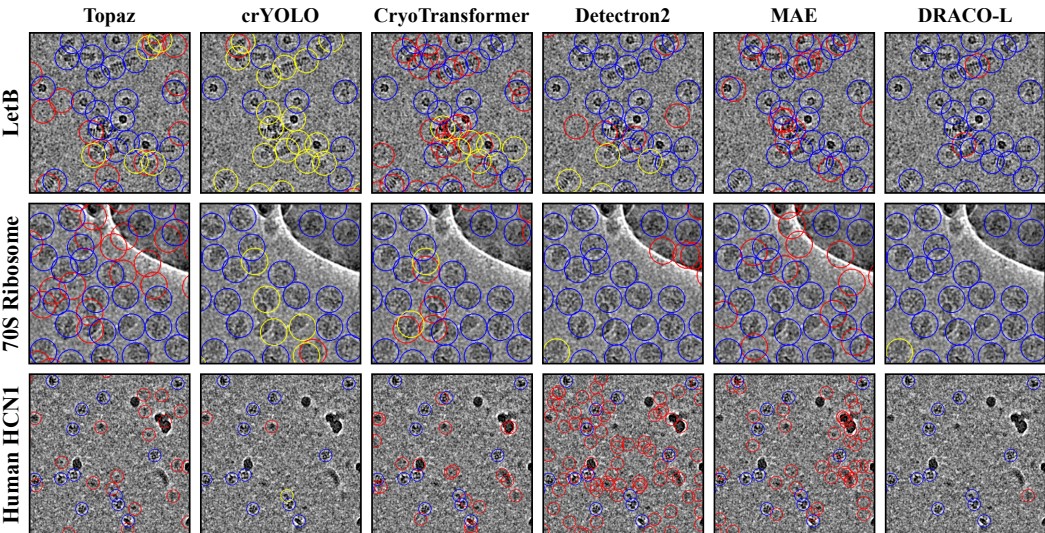

Figure 3: **Visualization of particle picking results.** We show the picking results of DRACO and baselines on the test datasets range from small transport proteins to huge ribosomes. Blue, red, and yellow circles denote true positives, false positives, and false negatives, respectively.

Table 1: **Particle picking results.** We report the precision, recall, F1 score, and resolution on each test dataset among all baselines. The resolution is obtained from the default cryoSPARC workflow. We compare DRACO with existing state-of-the-art methods and consistently achieve the best F1 score and resolution of reconstructed results.

| Method | Human HCN1 | | | | 70S ribosome | | | | LetB | | | |
|---|---|---|---|---|---|---|---|---|---|---|---|---|
| | Precision (↑) | Recall (↑) | F1 score (↑) | Res. (↓) | Precision | Recall | F1 score | Res. | Precision | Recall | F1 score | Res. |
| Topaz | 0.462 | **0.956** | 0.623 | 4.20 | 0.362 | **0.943** | 0.523 | 2.80 | 0.518 | 0.761 | 0.617 | 3.67 |
| crYOLO | **0.818** | 0.748 | 0.782 | 4.15 | 0.602 | 0.869 | 0.711 | 2.78 | 0.632 | 0.163 | 0.224 | 4.62 |
| CryoTransformer | 0.475 | 0.910 | 0.624 | 4.13 | 0.517 | 0.887 | 0.654 | 2.79 | 0.429 | 0.706 | 0.534 | 3.67 |
| Detectron | 0.392 | 0.834 | 0.533 | 4.50 | 0.668 | 0.901 | 0.767 | 2.85 | 0.589 | 0.804 | 0.680 | 3.86 |
| MAE | 0.703 | 0.649 | 0.675 | 4.32 | 0.712 | 0.876 | 0.786 | 2.84 | 0.591 | **0.805** | 0.682 | 4.03 |
| **DRACO-B** | 0.768 | 0.799 | 0.793 | 4.03 | 0.732 | 0.905 | 0.810 | 2.61 | 0.637 | 0.779 | 0.701 | 3.55 |
| **DRACO-L** | 0.830 | 0.802 | **0.816** | **3.90** | **0.803** | 0.846 | **0.824** | **2.51** | **0.678** | 0.780 | **0.725** | **3.53** |

produce 3D reconstruction density maps and the corresponding resolution, as described in Appendix C.3.

**Results.** As illustrated in Figure 3, both Topaz and CryoTransformer tend to pick a larger number of particles, but this often results in many false positives. In contrast, crYOLO achieves higher precision in picking, yet exhibits a higher number of false negatives. Detectron2 trained from scratch and pre-trained MAE both have difficulties in distiguishing signal and noise, leading to a lack of generalizability. In contrast, DRACO effectively identifies correct particles, surpassing the performance of all baselines, as demonstrated in Table 1.

## 5.2 Micrograph Denoising

Once pre-trained, our model can naturally serve as a generalizable denoiser by directly predicting every patch of input noisy micrograph without any further fine-tuning.

**Baseline and metrics.** To evaluate the effectiveness of DRACO on the denoising task, we first compare DRACO with the standard MAE trained on the 270,000 micrographs from our large-scale curated dataset with ViT-B as the backbone, denoted as **MAE**. We further compare with a popular denoising method **Topaz-Denoise** [37] in cryo-EM. For a fair comparison, we train Topaz-Denoise on our odd-even micrograph dataset for 100 epochs with default settings. Last, we compare with the traditional method **Low-pass** filtering that has already been integrated into commercial software cryoSPARC [41]. We utilize the same protocol used in cryoSPARC and set the low-pass cutoff resolution to 20 Å. As cryo-EM micrographs lack clean ground truth, following Topaz-Denoise, we employ an SNR calculation method that involves 20 manually annotated signal-background region pairs as references. For each $i$-th pair, we calculate the mean and variance for both the signal region

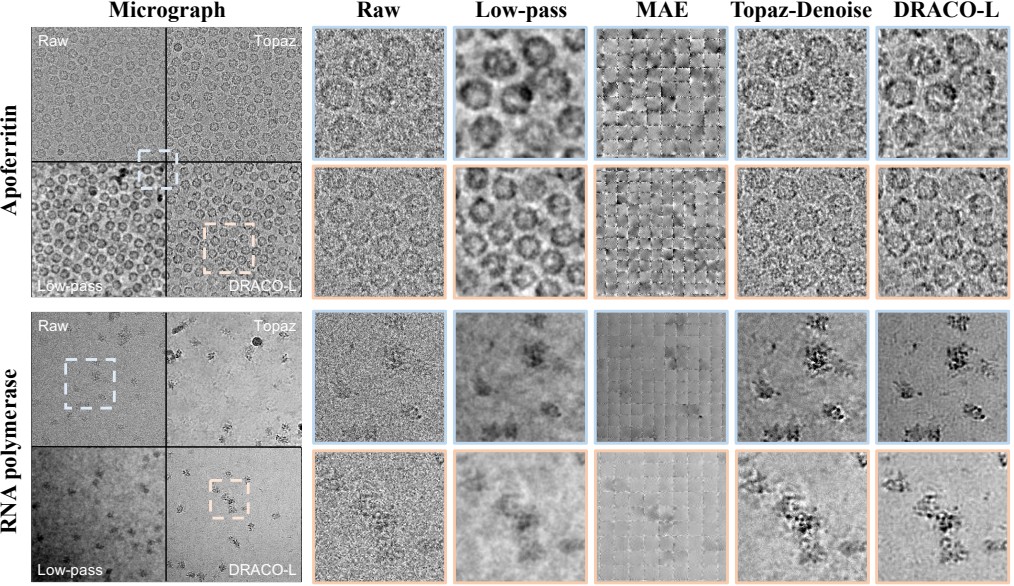

Figure 4: **Qualitative comparison results of micrograph denoising.** We visualize the denoising results of DRACO and state-of-the-art baselines. Our results show the most significant SNR improvement without the loss of the particle structure details. In contrast, Low-pass leads to a severe blur on particles, MAE introduces severe patch-wise artifacts and Topaz only shows either minor SNR improvements or blurred results.

Table 2: **Quantitative comparison of denoising results.** We report the SNR calculated with Equation 11. On our test dataset, DRACO outperforms all other baselines.

|  | Human Apoferritin | HA Trimer | Phage MS2 | RNA polymerase |
|---|---|---|---|---|
| Raw | -10.01 | -6.52 | -12.52 | -4.69 |
| Low-pass | -2.18 | -0.84 | -6.71 | 3.09 |
| Topaz-Denoise | -5.67 | -0.83 | -6.93 | 8.66 |
| MAE | -0.31 | -1.85 | -8.45 | 1.27 |
| **DRACO-B** | 1.92 | **3.69** | -0.13 | 10.13 |
| **DRACO-L** | **2.01** | 3.33 | **0.23** | 12.21 |

$r_i^s$ and the background region $r_i^b$, yielding $\mu_i^s$, $v_i^s$ for the signal, and $\mu_i^b$, $v_i^b$ for the background. The average SNR is then computed in dB as follows:

$$\text{SNR} = \frac{10}{N} \sum_{i=1}^{N} \log_{10} \frac{(\mu_i^s - \mu_i^b)^2}{v_i^b} \tag{11}$$

**Dataset.** To evaluate the denoising capabilities of the baseline models, we select four original micrograph datasets as test datasets, which are *excluded* from the training set. These sets are Human Apoferritin (EMPIAR-10421) [59], HA Trimer (EMPIAR-10096) [60], Phage MS2 (EMPIAR-10075) [61], and RNA polymerase (EMPIAR-11521) [62]. For each dataset, 5 micrographs are selected and 20 signal-background pair regions are labeled in total. Specifically, these signal and background regions are chosen close together to ensure similar background signals across both regions.

**Results.** The quantitative experiments show that DRACO achieves significant performance improvements in terms of SNR after denoising compared with state-of-the-art methods, as shown in Table 2. In Figure 4, the standard MAE can only recover smooth contours of the particle with severe artifacts. Low-pass filtering smooths both signal and background noise, but the background noise is still relatively high and the structure information of particles is corrupted. Topaz sometimes fails to effectively denoise micrographs but over-smooth them instead, which affects the generalizability. In contrast, DRACO outperforms all baselines in retaining the original particle signals with the lowest background noises, showing the best generalizability. We additionally reconstruct 3D density map and corresponding resolution using the denoised particles generated by each method. To ensure

Table 3: **Quantitative comparison of reconstruction using denoised particles.** We report the resolution obtained from the standard cryoSPARC workflow. On our test dataset, DRACO outperforms all other baselines.

|  | Human Apoferritin | HA Trimer | Phage MS2 | RNA polymerase |
|---|---|---|---|---|
| Low-pass | 2.63 | **2.06** | 3.46 | 2.75 |
| Topaz-Denoise | 2.34 | 3.06 | 2.52 | 2.93 |
| MAE | 2.77 | 2.15 | 3.78 | 2.81 |
| **DRACO-B** | **2.05** | 2.10 | **2.51** | **2.56** |

Table 4: **Quantitative comparison of micrograph curation.** Miffi employs its own general model, while ResNet18 is trained from scratch. DRACO reports the best results on all four classification metrics.

| Method | Accuracy | Precision | Recall | F1 score |
|---|---|---|---|---|
| Miffi | 0.836 | 0.899 | 0.845 | 0.871 |
| ResNet18 | 0.938 | 0.923 | 0.960 | 0.940 |
| MAE | 0.904 | 0.927 | 0.892 | 0.909 |
| **DRACO-B** | 0.963 | **0.976** | 0.953 | 0.964 |
| **DRACO-L** | **0.983** | **0.976** | **0.992** | **0.984** |

that comparisons reflect only the impact of denoising quality, we fix the locations and poses of the picked particles, which were determined using the cryoSPARC workflow. The results are shown in Table 3. DRACO consistently achieves the highest resolution in most cases, demonstrating its effectiveness in preserving more high-frequency signals while effectively reducing background noise. We demonstrate additional denoising results in Figure 5.

## 5.3 Micrograph Curation

A modern cryo-EM can capture thousands of micrographs in a day. However, the quality of captured micrographs is unverified. Low-quality micrographs may arise from artifacts such as empty sample, ice crystals, ethane contamination, severe drifting, etc. [40]. Low-quality micrographs can negatively contribute to the final reconstruction results. A reliable automated micrograph curation method can significantly improve the efficiency of the data processing pipeline, resulting in shorter processing time and improved final resolution. We show that DRACO can easily adapt to this 2-class classification task by linear probing, and achieving the best performance compared to the state-of-the-art method. Similar to [6], we freeze the encoder backbone and train an extra linear classification head.

**Dataset.** We manually annotate 1,194 micrographs from original micrograph datasets, assigning a binary label (accept or reject) to each to indicate quality. The dataset comprises 617 high-quality and 577 low-quality micrographs. We divided these micrographs into training and evaluation datasets using an 80%/20% split ratio.

**Baseline and metric.** We compare **DRACO-B** and **DRACO-L** with linear probing against several baselines: a small **ResNet18** [63]; an existing supervised method **Miffi** [40], which has been trained on 45,000 annotated data; and the standard **MAE** with linear probing both pre-trained and adapted on our datasets. The ResNet18 is trained from scratch to show the effectiveness of our pre-training strategy. More configuration details is provided in Appendix D.2. We report the widely used metrics for classification including precision, recall, F1 score, and accuracy on our test dataset. As Miffi predicts multi-labels on low-quality micrographs, we consider them all as rejections for our metric calculations.

**Results.** As shown in Table 4, Miffi, limited by insufficient training data, lacks generalizability on the test dataset. Both ResNet and MAE face difficulties in effectively separating noise from signal for classification for accurate classification. In contrast, DRACO extracts global information from noisy images more effectively, resulting in higher classification accuracy and demonstrating better generalizability compared to other methods.

## 5.4 Ablation study

We compare the performance of networks with different parameter sizes in particle picking and micrograph curation tasks, as shown in Table 1 and Table 4. The results demonstrate that our method

Table 5: **Evalution of mask ratios.** We demonstrate the performance of DRACO with different mask ratios. The result shows that at the 0.75 mask ratio, DRACO achieves the best performance.

| Mask Ratio | Micrograph Curation | | | | Denoising | |
|---|---|---|---|---|---|---|
| | Accuracy | Precision | Recall | F1 Score | RNA polymerase | HA Trimer |
| 0.5 | 0.954 | 0.968 | 0.945 | 0.956 | 10.29 | 2.57 |
| 0.625 | 0.930 | 0.960 | 0.909 | 0.933 | **10.49** | 2.80 |
| 0.75 | **0.963** | 0.976 | **0.953** | **0.964** | 10.13 | **3.69** |
| 0.875 | 0.958 | **0.984** | 0.939 | 0.961 | 9.59 | 2.28 |

Table 6: **Evalution of loss function.** We demonstrate the performance of DRACO with different training objective on the 70S ribosome dataset. The result shows that with both loss, DRACO achieves the best performance.

| Training scheme | Particle Picking | | | | Denoising |
|---|---|---|---|---|---|
| | Precision($\uparrow$) | Recall($\uparrow$) | F1 Score($\uparrow$) | Res.($\downarrow$) | SNR($\uparrow$) |
| **DRACO-B w/o N2N** | 0.712 | 0.876 | 0.786 | 2.84 | -4.94 |
| **DRACO-B w/o recon** | 0.713 | 0.817 | 0.761 | 2.85 | -4.22 |
| **DRACO-B** | **0.732** | **0.905** | **0.810** | **2.61** | **-2.86** |

can effectively scale up. Additionally, we evaluate the impact of different mask ratios on denoising and micrograph curation performance, as shown in Table 5. DRACO achieves the highest SNR and curation metric at a 0.75 mask ratio, thus we choose 0.75 as our default mask ratio. We have also conducted an additional ablation study of loss design. Specifically, we remove either the N2N loss (**w/o N2N**) or the reconstruction loss (**w/o recon**) from the training and evaluate the resulting models on particle picking and denoising tasks, as shown in Table 6. The result shows that both N2N and reconstruction losses improve performance. Without the N2N loss, DRACO struggles to distinguish between signal and noise in micrographs. Without the reconstruction loss, DRACO loses its ability to extract general features. We show additional visualization of their denoising results (Figure 6) in the Appendix A.

# 6 Discussion

**Limitations.** As the first attempt to achieve robust feature extraction for cryo-EM via a novel denoising-reconstruction autoencoder, our work presents opportunities for future enhancements. First, our method relies heavily on the performance of motion correction algorithms. This can be improved by designing a more comprehensive denoising task for raw noisy movies. Second, although we have collected what we believe to be the largest curated dataset for cryo-EM, it focuses primarily on mainstream single-particle datasets. To enhance dataset diversity, other types of cryo-EM datasets, such as cryo-electron tomography (cryo-ET)[64] datasets, should also be included. Finally, our current approach only supports various micrograph-level downstream tasks. For particle-level tasks, such as pose estimation, a more fine-grained yet robust feature extraction is required. This can be achieved by developing a particle-level version of DRACO.

**Conclusion.** We have introduced DRACO, a foundation model designed specifically for cryo-EM image processing, supported by a unique denoising-reconstruction pre-training framework to enable robust feature extraction for cryo-EM micrographs. We have constructed a diverse and high-quality cryo-EM image dataset from the uncurated public database, comprising over 270,000 movies and micrographs. After pre-training, our model's versatility is evidenced by its superior adaptation performance across multiple downstream tasks, including denoising, micrograph curation, and particle picking. All code, pre-trained model weights, and datasets will be made publicly available for further research and model development.

# 7 Acknowledgement

This work was supported by HPC Platform of ShanghaiTech University. We thank Zhenyang Xu for shaping the paper.

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

# A  Additional Results of DRACO

We demonstrate that DRACO achieves the highest visual denoising quality in terms of both signal preservation and noise removal, as shown in Figure 5. This figure serves as an extension to Figure 4 in our main paper. Furthermore, we visualize the denoising results from our ablation study on different mask ratios in Figure 6. We observe that at a 0.75 mask ratio, DRACO achieves the best denoising results, as supported by Table 2 in our main paper. Lastly, we visualize the reconstruction ability at a 0.75 mask ratio in Figure 7. DRACO demonstrates comparable reconstruction capability to MAE, while significantly better denoising results on visible patches.

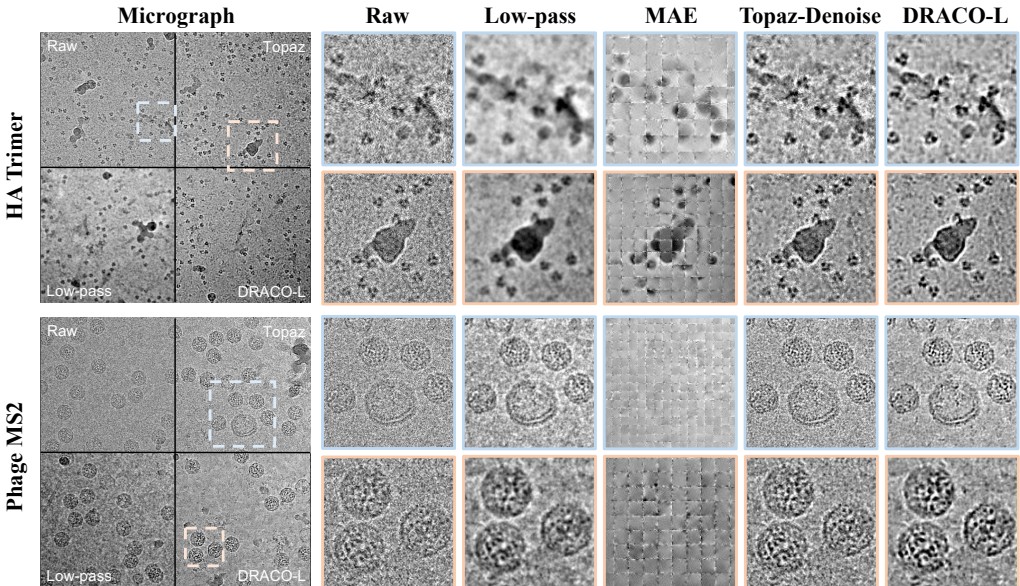

Figure 5: **Additional denoising results.** We have conducted additional experiments on datasets of membrane proteins and bacteriophages. DRACO achieves the highest visual denoising quality by optimally balancing signal preservation and noise reduction.

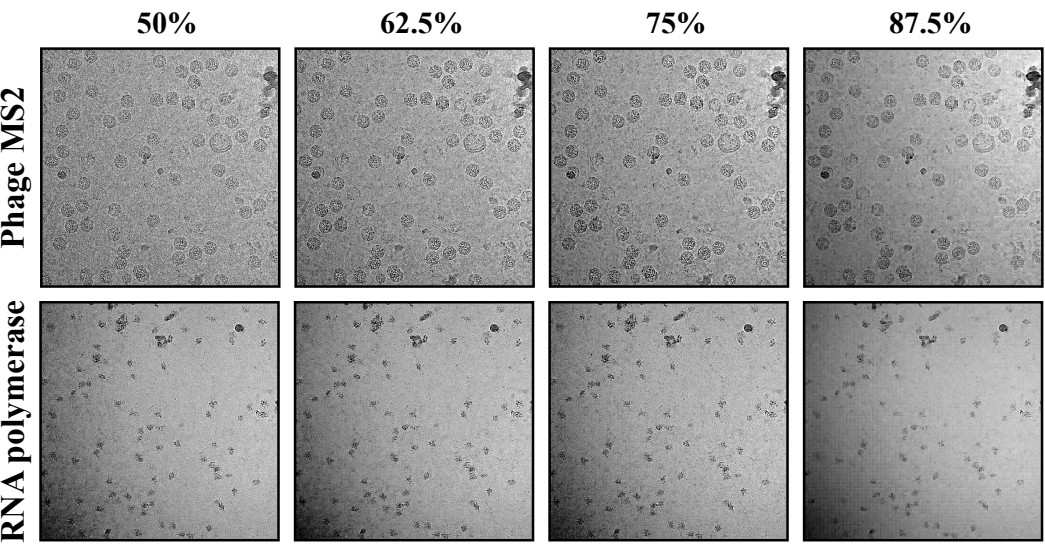

Figure 6: **Visualization of DRACO's results on different mask ratios.** At a 0.75 mask ratio, DRACO achieves the best trade-off between signal preservation and background noise removal.

| Raw | Masked | MAE | DRACO |
|-----|--------|-----|-------|

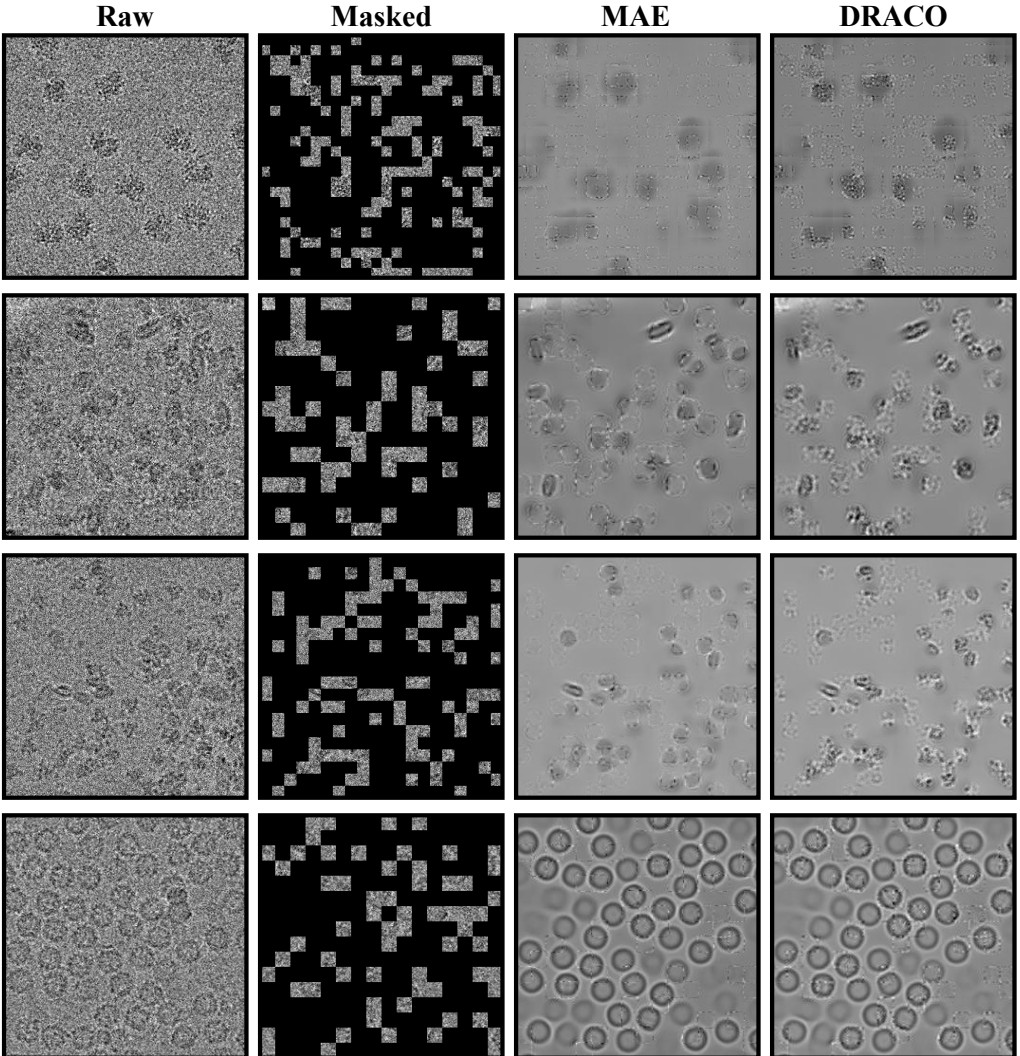

Figure 7: **Additional results on image reconstruction.** We present the reconstruction results at various image resolutions while maintaining a consistent mask ratio of 0.75. DRACO demonstrates enhanced detail preservation on the visible patches compared to MAE.

## B   Zero-shot Capability on Cryo-ET

Though DRACO has not been pre-trained on cryo-ET datasets, we found that DRACO can be directly applied on cryo-ET tilt series. Here, we demonstrate that DRACO is capable of denoising the unseen HIV tilt series [65], as shown in Figure 8. Specifically, we evaluate DRACO on both the tilt series and the volume slices, showing that DRACO effectively removes background noise and achieves higher contrast in the volume. Furthermore, we assess DRACO's performance in the context of reconstruction. We compare the slices from both tomograms reconstructed using original and denoised tilt series. Additionally, we compare these with the denoised slice from the original tomogram. The result shows that DRACO can improve the contrast of slices before or after the reconstruction.

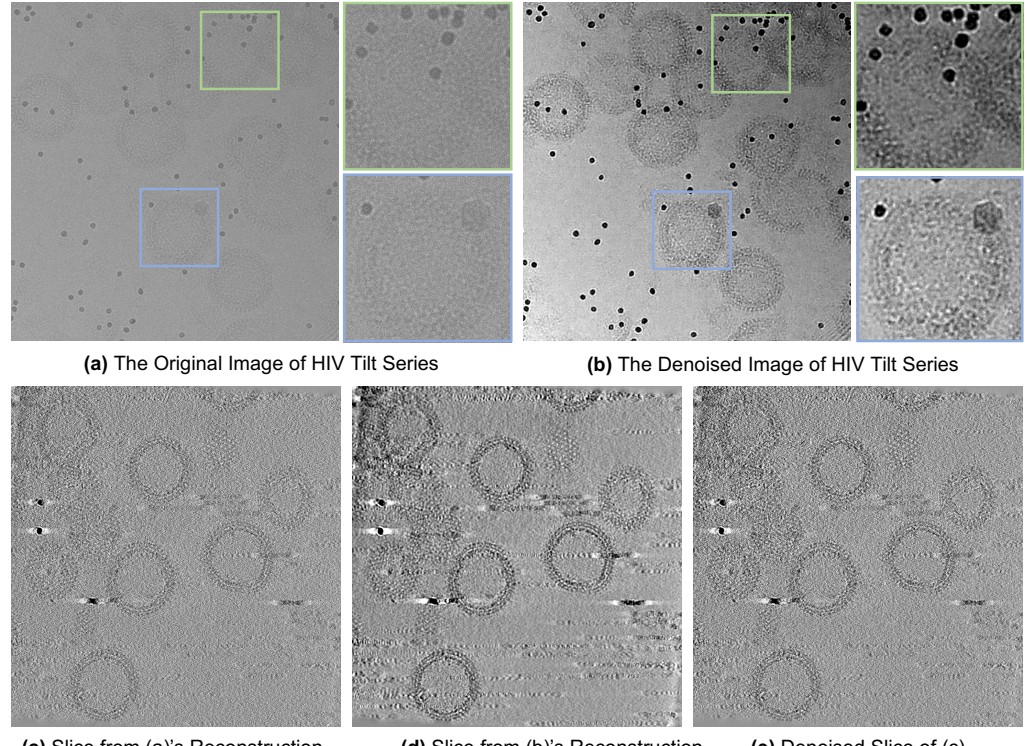

**(a)** The Original Image of HIV Tilt Series   **(b)** The Denoised Image of HIV Tilt Series

**(c)** Slice from (a)'s Reconstruction   **(d)** Slice from (b)'s Reconstruction   **(e)** Denoised Slice of (c)

Figure 8: **Denoising cryo-ET HIV tilt series with DRACO.** Figure (a) and (b) show the HIV tilt series beforeand after DRACO's denoising process. Using IMOD, we reconstruct 3D volumes of HIV from both the original and denoised series, showing their slice in Figures (c) and (d). Note the horizontal stripes in these images, which are artifacts due to the missing wedge issue in cryo-ET. Figure (e) shows a denoised slice from Figure (c) by DRACO.

## C   Workflow of CryoSPARC

### C.1   Pre-training Dataset Details

EMPIAR [17] is a public archive for storing raw cryo-EM images and 3D reconstructions from vEM and XT experiments. It currently contains over 2,000 entries, totaling more than 2 PB of data. The EMDB [66] is an archive of 3D reconstructions derived from cryo-EM experiments, many of which supplement the experimental information of the EMDB-related EMPIAR entries, such as sample preparation and reconstruction processes.

In the field of 3D electron microscopy, cryo-EM micrographs undergo complex image processing to achieve 3D reconstructions with specified resolutions. Experts can model protein molecules accurately on 3D reconstructions with resolutions better than 3 Å. We define the quality of datasets from the perspective of structural biology; hence, high-quality datasets should produce high-resolution 3D reconstructions suitable for detailed analysis.

When constructing the pre-training dataset, we utilize the REST API provided by EMPIAR to obtain metadata for each dataset, such as experiment type, EMDB ID, and image classification. We specifically filter for EMPIAR entries that are experimentally linked to EMDB, prioritizing those with reconstruction resolutions better than 10 Å. Subsequently, we collect as many datasets as possible that contain single-frame micrographs. For datasets that include multiple frames, we further process them by separating the frames into odd and even micrographs.

Cryo-EM raw data consists of multi-frame recordings known as movies, which capture the number of electrons. After motion correction [13], these movies yield single frames referred to as micrographs. We download as many single-frame micrographs as possible, categorized by image type. Additionally,

we process multi-frame movies into single-frame micrographs using cryoSPARC's Patch Motion Correction. For each multi-frame movie, we separately process the complete frames, odd frames, and even frames to generate three types of single-frame micrographs for DRACO learning.

## C.2  Particle Picking Dataset Details.

We filter and generate a dataset comprising approximately 80,000 single-frame micrographs annotated with about 8 million particles, following a process based on the cryo-EM single particle analysis reconstruction pipeline software, cryoSPARC [41]. Each EMPIAR public dataset [17] comes with a solved 3D density map, available on EMDB [66]. Using the "create template" step in cryoSPARC, we project this map into 50 diverse poses to generate high-quality templates for template picking. Subsequent rounds of the "2D classification" step are employed to eliminate potential false positives. Finally, using these particles, we reconstruct results whose resolution did not differ by more than 20% from the reported resolution. This method allowed us to collect a high-quality annotated particle dataset.

## C.3  3D Reconstruction Pipeline

The reconstruction process is also based on cryoSPARC. After picking the particles, the standard reconstruction workflow consists of 2D classification, ab initio reconstruction and homogeneous refinement. 2D classification aims to remove any false positives in picked particles. Ab initio reconstruction can create an initial 3D model from a certain set of particles. Based on this initial model, homogeneous refinement can achieve a high-resolution result. The final resolution is determined by the Fourier shell correlation (FSC) curve. The specific method involves dividing the particles into two random halves, each undergoing homogeneous reconstruction. After reconstruction, we perform a cross-correlation on each Fourier shell in the frequency domain of two reconstructed 3D density maps. The final resolution is determined using the standard threshold of 0.143 on the FSC curve.

# D  Downstream tasks settings

## D.1  Particle Picking Settings

**Particle picking baselines.**    We use the Topaz [51] general model with its "resnet16u64" backbone for our baseline, picking particles that score higher than 0.0 as the final results. We use the crYOLO [52] general model for our baseline from its official website. Similarly, we use CryoTransformer's [53] open-source general model on  Github, choosing particles with scores ranging from the 25th to the 100th percentile.

**Detectron2 configurations.**    For particle picking, we employ the Faster R-CNN framework within Detectron2, to fit the particle picking task with our curated dataset. The configurations for both ViT-B and ViT-L include the standard feature pyramid and window attention [67]. We also set the non-maximum suppression threshold of the region proposal network to 0.6 and adjust the pooling size of the box pooler in the region of interest network to 14. Given that the particles are mostly square, the aspect ratio of the anchors is fixed to 1.0. All other settings remain at their default values. The data augmentation goes the same process as described in the pre-training stage. We fine-tune Detectron2-based particle picking model on 64 NVIDIA A800 GPUs for 100 epochs with a batch size of 256, requiring approximately 9 hours and consuming around 100GB of memory. After fine-tuning, we process the test dataset and pick particles with scores higher than 0.1 as the final results.

## D.2  Micrograph Curation Settings

For micrograph curation, each ViT-based model undergoes a linear probing phase, while a ResNet18 [63] is trained from scratch. We employ the miffi_v1 [40] general model of Miffi to inference on test datasets. All the model trains 50 epochs with a batch size of 128 on a single NVIDIA RTX 3090 GPU, taking about 10 minutes and utilizing around 8GB of memory.

