# OpenReview forum: "DRACO: A Denoising-Reconstruction Autoencoder for Cryo-EM"
_NeurIPS.cc/2024/Conference — NeurIPS 2024 poster_

### Official Review · Reviewer_AE1Z · 2024-06-24

**Soundness:** 3
**Presentation:** 3
**Contribution:** 3
**Rating:** 6
**Confidence:** 5

**Summary:**

This manuscript introduces a new denoising autoencoder for cryo-EM micrographs based on vision transformers (ViTs) which is trained on masked pairs of images with a noise2noise loss function paired combined a reconstruction loss. In addition to the model itself, an important contribution of the work is the curation of a large dataset of experimental micrographs used to train the model, containing a total of 270 000 micrographs from publicly available sources. The resulting model is shown to achieve state-of-the-art performance on denoising, particle picking, and micrograph curation.

**Strengths:**

The manuscript describes the problem to be solved accurately and outlines why traditional methods for training autoencoders do not perform well. The overall description of the proposed model is also well illustrated through the use of figures and well-defined notation. The considerable effort expended to ensure that the training and testing datasets are of sufficient quality is also a major strength of the work. Finally, the results on the three evaluation tasks are quite impressive, even though some questions arise regarding their calculation.

**Weaknesses:**

The manuscript is lacking in details in several important areas. First, the description of the method is sometimes at a very superficial level, making it hard to understand how it is actually implemented. For example, the mask is given as input to the decoder in eq. (5), but it is not clear how this is given as an input to the ViT. Furthermore, the [MASK] token is simply described as a “shared learnable embedding” without any more details. Clearly spelling out the details of the architecture in the main text of in an appendix would significantly strengthen the work. This also goes for the adaptation of the network to the other two downstream tasks (particle picking and micrograph curation), where the authors simply state that they “conduct supervised fine-tuning based on Detectron2” (for particle picking) and that they “freeze the encoder backbone and train an extra linear classification head” (for micrograph curation).

Second, the motivation for the various choices made is sometimes missing. This goes for the choice of architecture but also for the construction of the loss function, the choice of patch size, data augmentation strategy, and so on.

Third, the evaluation methods for the first two task need to be better motivated. For example, the SNR calculation method in eq. (7) is simply stated (with a reference to the Topaz-denoise paper) without further explanation. Similarly, the definition of true and false positives and false negatives for the particle picking task needs to be spelled out.

The final weakness is Section 5.4, which purports to be an ablation study. It refers to different results based on parameter sizes in Tables 2 and 3, but those results do not seem to be there. Furthermore, this section seems to be more of a hyperparameter study, rather than an ablation study (no steps of the pipeline are removed or replaced by others here).

There are also some issues with language, such as the reference to “theological analysis” on page 4 and “a modern cryo-EM” and “artifacts such as empty” on page 8.

**Questions:**

– Why use a ViT as opposed to a simpler convolutional network?
– How does masking an image make sense on images such as micrograph which typically lack long-range structure?
– Please define γ on page 4.
– Why is the reconstruction target necessary for the proper training of the network? How does the dependence of the noise realization between input and target affect training here.
– How do we expect the mask ratio to affect the results? Why does this trade off between “signal preservation and background noise removal”?

**Limitations:**

The authors have largely addressed the limitations of the work in the manuscript.

---

> ### Author Rebuttal · Authors · 2024-08-06
>
> # Reviewer AE1Z (5)
>
> We sincerely thank the reviewer for the detailed review of our work.
>
> ## On the Clarification of Details in the Method
>
> We truly appreciate your detailed suggestions for our method. Here, we clarify the details in below:
>
> 1. Similar to MAE [He K et al, 2022], the mask ratio $\gamma$ is a hyperparameter that represents the percentage of masked patches in the patch set.
> 2. In Eq. 5, we use the binary mask $\\{\mathbf{m}_i\\}\_\{i=1\}^N$ to select visible patches (where $\mathbf{m}_i = 1$) from input patch sets. We will make it clearer in our revision.
> 3. [MASK] token is a high-dimensional learnable embedding, whose dimension is 768 for DRACO-B and 1024 for DRACO-L. As described in Eq. 6, it is used to pad the masked image tokens.
>
> We will make all these points clear and add more details on DRACO's architecture in our revision.
>
> ## On the Clarification of Design Choices
>
> Thank you for your constructive suggestions.
>
> **Design choice of ViT-based DRACO.** We use ViT [Dosovitskiy A et al, 2020] as the backbone for DRACO, following MAE [He K et al, 2022]. We choose ViTs rather than CNNs due to their scalability and effectiveness in learning from large datasets [Dosovitskiy A et al, 2020]. For the long-range structure lacking issue in DRACO pre-training, as the particle size in the training dataset ranges from 100 to 200, they can serve as long-range structures compared with 16 x 16 visual token size. For data augmentation, given the high resolutions of raw micrographs (4096 x 4096), we randomly crop patches from the original micrograph with side lengths ranging from 1/16 to 1/4 of the original size and rescale them to 256 x 256, to reduce the memory and computation consumption.
>
> **Particle picking adaption.** We utilize the ViTDet [Li Y et al, 2022] architecture within the Detectron2 framework for particle picking, starting by loading DRACO's pre-trained encoder weights into Detectron2’s backbone for further fine-tuning. During adaption, a raw micrograph serves as input, with the encoder extracting a sequence of image tokens. A feature pyramid network with four parallel CNNs generates a multi-resolution feature map. This map feeds into a region proposal network, which predicts potential particle locations. Subsequently, a region of interest network refines these predictions into binary logits (particle or background) and more accurate locations. Specifically, we adapted the cross-entropy loss for 2-class classification in cryo-EM, with default training strategy and other loss functions. We will add these detailed descriptions to the revision.
>
> **Micrograph curation adaption.** We utilize DRACO's encoder as the feature extractor, coupled with a single linear layer that outputs 2-class logits indicating the probability of a micrograph being good (1) or bad (0). We employ binary cross-entropy loss to supervise this predicted value. The encoder weights remain frozen during this linear probing adaption.
>
> We will make all these details clearer in the revision.
>
> ## On the Clarification of Metrics and Notations
>
> Thank you for thoroughly reading our paper, and we truly appreciate your suggestions. We will clarify below points in the revision.
>
>
> We follow the state-of-the-art denoising method, Topaz-Denoise [Bepler T et al, 2020] to manually create particle-background pairs to calculate SNR metric for the evaluation of denoising tasks. SNR is a measure of the signal strength to the background noise that has been widely used for measuring the quality of the image in cryo-EM.
>
> We follow Topaz [Bepler T et al, 2019] to use true positive, false positive, and false negative to represent a correctly detected particle, a wrongly detected particle, and an undetected particle, respectively.
>
> ## On the Ablation Study
>
> In the ablation study section, we followed the practices of MAE [He K et al, 2022] for parameter size and mask ratio.
>
> **Ablation study of parameter size.** In Tables 2 and 3, DRACO-B and DRACO-L refer to taking ViT-B (86M parameters) and ViT-L (307M parameters) as the encoders used on downstream tasks, respectively. The results show that a larger backbone leads to slightly better performance on downstream tasks overall. We will clarify the parameter size differences in the revision.
>
> **Ablation study of the mask ratio.** In DRACO, the mask ratio determines the proportion of the N2N loss and the reconstruction loss. This influences the network's capacity to capture both high and low-frequency signals in micrographs. As shown in Table 4 of our paper, the practically best mask ratio is 0.75 to reasonably remove background noise while preserve high-frequency signals.
>
> **Ablation study of the loss functions.**  As suggested, we have conducted an additional ablation study of loss design. In the following table, we remove either the N2N loss (**w/o N2N**) or the reconstruction loss (**w/o recon**) from the training and test the variations on particle picking and denoising tasks. The result shows that both N2N and reconstruction losses improve performance.
>
> |Task|Metric|DRACO-B w/o N2N|DRACO-B w/o recon|DRACO-B|
> | - | - |:-:|:-:|:-:|
> |**Particle Picking**|**Precision ($\uparrow$)**|0.712|0.713|**0.732**|
> ||**Recall ($\uparrow$)**|0.876|0.817|**0.905**|
> ||**F1 score ($\uparrow$)**|0.786|0.761|**0.810**|
> ||**Res. (Å, $\downarrow$)** |2.84|2.85|**2.61**|
> |**Denoising**|**SNR ($\uparrow$)**|-4.94|-4.22|**-2.86**|
>
> If space permits, we will conduct the ablation study in terms of patch size and the data augmentation strategy in the revision.

---

> > ### Comment · Reviewer_AE1Z · 2024-08-13
> >
> > Thank you for your rebuttal. Given the changes you propose, I will raise my score to 6.

---

### Official Review · Reviewer_b2KB · 2024-07-07

**Soundness:** 2
**Presentation:** 2
**Contribution:** 2
**Rating:** 4
**Confidence:** 2

**Summary:**

In this paper, the authors introduce DRACO, a Denoising-Reconstruction Autoencoder for Cryogenic Electron Microscopy (cryo-EM) inspired by the Noise2Noise (N2N) approach. DRACO aims to address the high-level noise corruption in cryo-EM images, which are often overlooked by other foundation models in computer vision. The method involves processing cryo-EM movies into odd and even images, treating them as independent noisy observations, and applying a denoising-reconstruction hybrid training scheme. By masking both images, DRACO creates denoising and reconstruction tasks to enhance the quality of the output.

For the pre-training phase, the authors build a high-quality and diverse dataset from an uncurated public database, comprising over 270,000 movies or micrographs. This dataset is essential for ensuring the effectiveness of DRACO as a generalizable cryo-EM image denoiser and a foundation model for various downstream tasks. The experimental results demonstrate that DRACO outperforms state-of-the-art baselines in denoising, micrograph curation, and particle picking tasks.

**Strengths:**

The paper is well-organized and clearly written, with a logical flow of information from the problem statement to the proposed solution and experimental validation. The authors provide a clear explanation of their denoising-reconstruction hybrid training scheme, including the masking of images to create denoising and reconstruction tasks, which enhances the reader’s understanding of the methodology.

**Weaknesses:**

Weaknesses
While the paper makes a valuable contribution, there are several areas for improvement:

Training Method Justification: The authors use a Noise2Noise (N2N) approach for training but do not justify why a Noise2Clean (N2C) approach was not considered, given the Poisson-Gaussian noise model in electron microscopy images.

Comparison with Blind-Spot Networks: The paper lacks a comparison with existing blind-spot network-based denoising methods. Including such a comparison would provide a clearer benchmark for evaluating the effectiveness of the proposed method.

Evaluation Metrics: The use of SNR as the sole evaluation metric may not fully capture the quality of restored images, as SNR tends to favor smoother results and may overlook important texture details.

**Questions:**

[Why Not Use N2C for Network Training]
The authors mention in the paper that the noise model for electron microscopy images is a Poisson-Gaussian noise model. Given this, why not use a Noise2Clean (N2C) approach by inputting simulated noisy images and learning the corresponding clean images for network training?

[Comparison with Blind-Spot Network-Based Methods]
There are existing denoising methods for real noisy images based on blind-spot networks. Why can't these methods be applied to electron microscopy image denoising? Furthermore, in the comparison with other methods, the authors only compare with low-pass filtering and MAE methods, making it difficult to clearly see the effectiveness of the proposed method.

[Is SNR Sufficient to Evaluate Restored Image Performance?]
The authors compare the SNR of restoration results from different methods. However, considering the calculation method of SNR, it may favor smoother restoration results and potentially overlook some texture details.

**Limitations:**

Can the authors construct a more novel paired electron microscopy image dataset?

---

> ### Author Rebuttal · Authors · 2024-08-06
>
> We thank the reviewer for their questions and suggestions on denoising. Based on this, we will further clarify the questions related to denoising baseline selection and metric evaluation.
>
> ## On the Choice of Denoising Baselines
>
> **Noise2Clean baseline.** Supervised image denoising methods like Noise2Clean (N2C) are prevalent in the field of computer vision, where generating or simulating clean-noisy pairs is typically feasible. However, Topaz-Denoise has pointed out that real clean micrographs are unavailable in cryo-EM. Therefore, we use a Noise2Noise (N2N) approach to make DRACO robust to noise. In theory, one could leverage simulation techniques to produce clean-noisy image pairs in cryo-EM for N2C training. However, the current state-of-the-art methods, such as InsilicoTEM [1], still suffer from inaccurate simulations and high computational costs. We will include a detailed discussion of this in the introduction.
>
> **Blind-Spot baseline.** We appreciate the reviewer's suggestion. The Blind-Spot method is a reasonable baseline for denoising without clean data. We trained and evaluated its performance, and the results, shown in the following table, demonstrate that DRACO better distinguishes signal from background noise compared to the Blind-Spot method Blind2Unblind (B2U) [2]. For B2U, we apply the default settings and adapt them to our datasets. DRACO leverages the continuous multi-frame nature of cryo-EM images, allowing it to differentiate signal from noise more effectively than solely from noisy images. The MAE training paradigm also helps capture low-frequency signals, while Noise2Noise loss enables DRACO to focus on high-frequency signals.
>
> **Table 1**: The comparison of Blind-Spot and DRACO-B.
> |        Dataset        |   Metric   | Blind-Spot |  DRACO-B  |
> | --------------------- | ---------- | :--------: | :-------: |
> | **Human Apoferritin** |    SNR (dB, $\uparrow$ )     |   -5.98    | **1.92**  |
> |                       | Resolution (px.,  $\downarrow$) |    2.53    | **2.05**  |
> |     **HA Trimer**     |    SNR     |   -3.16    | **3.69**  |
> |                       | Resolution |    2.76    | **2.10**  |
> |     **Phage MS2**     |    SNR     |   -7.35    | **-0.13** |
> |                       | Resolution |    3.35    | **2.51**  |
> |  **RNA polymerase**   |    SNR     |   -1.63    | **10.13** |
> |                       | Resolution |    2.68    | **2.56**  |
>
> [1] Vulović M, Ravelli R B G, van Vliet L J, et al. Image formation modeling in cryo-electron microscopy[J]. Journal of structural biology, 2013, 183(1): 19-32.
>
> [2]Wang Z, Liu J, Li G, et al. Blind2unblind: Self-supervised image denoising with visible blind spots[C]//Proceedings of the IEEE/CVF conference on computer vision and pattern recognition. 2022: 2027-2036.

---

### Official Review · Reviewer_ypYy · 2024-07-08

**Soundness:** 3
**Presentation:** 2
**Contribution:** 3
**Rating:** 7
**Confidence:** 4

**Summary:**

In this paper, the authors introduce an autoencoder-based model to denoise cryo-EM micrographs. By splitting aligned movies into even and odd images, the authors obtain two images with similar signal and two different realizations of noise. Following Noise2Noise and Topaz-Denoise, the denoising target is to predict the "even" realization from the "odd" on and vice versa. The authors show that their method outputs micrographs with higher SNRs (as defined by Eq 11). The representation learned by the encoder can be used to fine-tune an object detection algorithm for particle picking or a linear layer for filtering micrographs.

**Strengths:**

**Large-scale curated dataset.** The authors give a detailed description of the protocol they followed to preprocess the EMPIAR datasets and pretrain their model.

**Description of the model.** The authors give a detailed description of the architecture of the model.

**Downstream model adaptation.** I find the possibility of using DRACO for model adaptation particularly exciting and potentially impactful. The authors illustrate this capability with two examples: particle picking and micrograph cleaning.

**Weaknesses:**

**Denoising section.** Section 5.1 describes the method as a denoising method. I found this section slightly misleading. To me, the main contribution of DRACO is its ability to learn a compact representation of cryo-EM patches, which can be used for downstream tasks. When the method is first described as a denoiser, the reader can be misled and think that downstream tasks (e.g particle picking) use the denoised images. Furthermore, I find that the evaluation of the denoising performances lacks quantitative analysis. Table 1 only reports the SNR obtained with a heuristic definition of the SNR, but reporting the resolution obtained after homogeneous reconstruction on the denoised images would be more compelling. If the authors think the denoising capability is *not* the main contribution of the paper, my recommandation would be to move down Section 5.1 to avoid misunderstandings. Otherwise, Section 5.1 should contain further quantitative evaluations.

**Questions:**

Do the authors plan to grant access to their pretraining datasets?

**Limitations:**

The limitations are well described at the end of the paper and the authors suggest interesting avenues for future work.

---

> ### Author Rebuttal · Authors · 2024-08-06
>
> We appreciate the reviewer's suggestions regarding the writing of the article. We will carefully consider your feedback and make the necessary improvements in the revision.
>
> ## Clarification of Denoising Section
>
> We apologize for the confusion and re-structure the paper. The reviewer is correct that denoising is one of the downstream tasks of DRACO, and the denoised results are not utilized in other downstream tasks.  As suggested, we will move this section further down in the paper to clarify.
>
> ## Dataset Release
>
> We are preparing to release our pre-trained dataset, which includes 529 sets of cryo-EM single-particle image data and their metadata, over 270,000 images in total. We will provide a public data server for file downloading.

---

> > ### Comment · Reviewer_ypYy · 2024-08-12
> >
> > I acknowledge that I have read the authors' rebuttal and the other reviewers's comment. My concerns have been addressed and I have decided to raise my score.

---

### Official Review · Reviewer_RjVX · 2024-07-08

**Soundness:** 3
**Presentation:** 3
**Contribution:** 3
**Rating:** 7
**Confidence:** 4

**Summary:**

This work proposes a foundation model for Cryo-EM tasks. The technique consists of pretraining a masked autoencoder on a curated dataset, with a Noise2Noise-like self-supervision scheme. The learned representations are then fine-tuned for various tasks, such as denoising, particle picking and micrograph curation. Experimental results show promise across all tasks compared to existing techniques.

**Strengths:**

- A foundation model for cryo-EM is an interesting contribution that has potential in a wide range of downstream applications in structural biology.
- The proposed training scheme is interesting and novel to the best of my knowledge.
- Authors curate a pretraining dataset which, if released to the community, can greatly benefit data-driven research for cryo-EM.
- The experimental results are promising.

**Weaknesses:**

- Justification for the training scheme is somewhat unclear. Why is the supervision signal from the original micrographs not used in every patch. By only utilizing the original micrograph on the masked area we discard some information. Alternatively, odd/even micrographs could be used to predict the original averaged micrograph.

- It is unclear how the method generalizes to unseen real data, a common scenario where foundation models are deployed. It would be important to understand how well the technique can be adapted in such cases.

- It seems like authors introduce a noise model (Gaussian + Poisson), but does not leverage the model further, beyond the zero-mean assumption.

**Questions:**

- How well would the method perform outside of the single-particle setting, for instance slices of cryo-ET?
- Can the proposed technique be adapted to 3D data such as that in cryo-ET?
- Typo: line 134 theological should be theoretical
- Vector notation would be better in Eq (8) and (9) (using norms)

**Limitations:**

Limitations are addressed.

---

> ### Author Rebuttal · Authors · 2024-08-06
>
> Thanks for your appreciation and insighful comments.
>
> ## Clarification of Supervision Target
>
> We thank the reviewer suggests that the supervision signal of visible patches can be replaced by the original micrograph to fully ultilize the high-quality information. In our paper, we followed the practice of Topaz-Denoise [1] to use odd/even micrographs as supervision signal. However, replacing them with the original micrographs would violate the Noise2Noise (N2N) assumption that noisy pairs are **independent** since the odd-original and even-original pairs are **dependent**. This dependency arises because the original micrographs share the same odd and even frames used in the generation of odd-even-original micrograph triplets.
>
> If space permits, we will include an ablation study that uses full micrographs to supervise the visible patches.
>
> [1] Bepler T, Kelley K, Noble A J, et al. Topaz-Denoise: general deep denoising models for cryoEM and cryoET[J]. Nature communications, 2020, 11(1): 5208.
>
> ## Fully Leveraging Noise Modeling
>
> The reviewer is correct that by far we have mainly exploited the zero-mean noise assumption. We'd emphasize that the Poisson-Gaussian noise model of cryo-EM follows the first principles [1], and the zero-mean conclusion is the theoretical basis for our use of Noise2Noise. Theoretically, we can further deploy the model to enhance 3D reconstructions via simulations schemes such as STEM [2]. We will include a detailed discussion in our revision.
>
> [1] Vulović M, Ravelli R B G, van Vliet L J, et al. Image formation modeling in cryo-electron microscopy[J]. Journal of structural biology, 2013, 183(1): 19-32.
>
> [2] Kniesel H, Ropinski T, Bergner T, et al. Clean implicit 3d structure from noisy 2d stem images[C]//Proceedings of the IEEE/CVF Conference on Computer Vision and Pattern Recognition. 2022: 20762-20772.

---

> > ### Comment · Reviewer_RjVX · 2024-08-11
> > **Response to rebuttal**
> >
> > I thank the authors for addressing my questions. I maintain my opinion that this work is an interesting contribution and keep my score.

---

### Author Rebuttal · Authors · 2024-08-06

# Global Response

**Please see the uploaded PDF for additional DRACO results on unseen cryo-ET data and a comparison with Blind2Unblind.**

We are encouraged that all reviewers recognize DRACO as a potentially beneficial foundation model adaptable to a broad range of cryo-EM downstream tasks. We thank the reviewers for their constructive suggestions, which we will incorporate into the revision. Below, we first address the common question raised by the reviewers and then respond to their individual comments.

## Generalization Capability of DRACO (**RjVX**)

We thank the reviewer for pointing out the potential application of DRACO in cryo-ET. We have conducted an experiment and included the results in the uploaded PDF (see Fig. 1). In this experiment, we demonstrate DRACO's zero-shot denoising capability on an unseen HIV cryo-ET tilt series [1]. Specifically, we evaluate DRACO on both the tilt series and the volume slices, showing that DRACO effectively removes background noise and achieves higher contrast in the volume. If space permits, we will include one such result in the revision.

We are excited to see that DRACO shows promising preliminary results on unseen cryo-ET data. In the future, we plan to incorporate the cryo-ET dataset into DRACO's training dataset and extend DRACO to the cryo-ET 3D volumes using the following approach. We will split the cryo-ET tilt series into odd and even tilt series and then reconstruct each to obtain odd/even 3D volume pairs. By training DRACO on corresponding slices from these paired volumes, we aim to achieve effective denoising for cryo-ET 3D volumes.

We'd also like to clarify that all experiments conducted and included in the paper were on unseen datasets, demonstrating the high generalizability of DRACO. We will make this point clearer in the revision.

[1] Schur, Florian KM, et al. "An atomic model of HIV-1 capsid-SP1 reveals structures regulating assembly and maturation." Science 353.6298 (2016): 506-508.

## On the Denoising Metrics (**ypYy**, **b2KB**)

We thank the reviewers point out that the Signal-to-Noise Ratio (SNR) may not comprehensively reflect the quality of restored images. Initially, we adopted SNR as the primary quantitative metric, following the example set by Topaz-Denoise [1]. Responding to the reviewers' recommendations, we have now expanded our evaluation to include experiments that assess particle reconstruction performance across various denoising methods (**ypYy**).

Specifically, we compare the reconstruction resolution across four datasets featured in our denoising section using various denoising schemes. To ensure that comparisons reflect only the impact of denoising quality,  we fix the locations and poses of the picked particles, which were determined using the cryoSPARC [2] workflow. We employ the Filtered Back Projection (FBP) algorithm for homogeneous reconstruction to determine the pixel resolution at an FSC of 0.143 for the two half-maps. As shown in the following table, DRACO consistently achieves the highest resolution in most cases, demonstrating its effectiveness in preserving more high-frequency signals while effectively reducing background noise.

We have included visualizations of the denoising results in the uploaded PDF. For additional comparisons, please see Fig. 2, which shows the Blind-Spot denoising method (**b2KB**).

**Table 1**: The comparison of the resolution (px.) $\downarrow$  of reconstructed volume by various baselines.

|Dataset|Low-pass|Topaz-Denoise|MAE|DRACO|
|-|:-:|:-:|:-:|:-:|
|**Human Apoferritin**|2.63|2.34|2.77|**2.05**|
|**HA Trimer**|**2.06**|3.06|2.15|2.10|
|**Phage MS2**|3.46|2.52|3.78|**2.51**|
|**RNA polymerase**|2.75|2.93|2.81|**2.56**|

[1] Bepler T, Kelley K, Noble A J, et al. Topaz-Denoise: general deep denoising models for cryoEM and cryoET[J]. Nature communications, 2020, 11(1): 5208.

[2] Punjani, Ali, et al. "cryoSPARC: algorithms for rapid unsupervised cryo-EM structure determination." Nature methods 14.3 (2017): 290-296.

---

### Comment · Area_Chair_3tSC · 2024-08-10

Dear reviewers,

Could you please respond to the rebuttal and discuss with authors?

---

### Decision · Program_Chairs · 2024-09-25

**Decision:**

Accept (poster)

**Comment:**

This work introduces DRACO, a foundation model for cryo-EM tasks. The model uses a hybrid denoising-reconstruction training strategy, with a Noise2Noise loss function paired with a reconstruction loss, to enhance the quality of cryo-EM images. The learned representations are fine-tuned for various downstream tasks, such as denoising, particle picking, and micrograph curation. Experimental results show that DRACO outperforms state-of-the-art techniques across these tasks, achieving higher signal-to-noise ratios (SNRs) and providing a robust solution for cryo-EM image processing. Although the reviewers pointed out some minor weaknesses (such as the motivation of choices, ablation studies), the rebuttal addresses most of them.